# The COVID-19 and chloroquine infodemic: Cross-sectional observational study of content analysis on YouTube

**Cynthia Poncelet** [1], **Raphaël Porcher**[2], **Yên-Lan Nguyen** [1]*

**1** Department of Anesthesiology and Intensive Care, Cochin Hospital, Paris, France, **2** Centre of Research in Epidemiology and Statistics (CRESS-UMR1153), National Institute of Health and Medical Research, Paris, France

\* yen-lan.nguyen@aphp.fr

**Data Availability Statement:** All relevant data are within the paper and its Supporting Information files.

**Funding:** The authors received no specific funding for this work.

## Abstract

The objective of this study is to evaluate the association between quality and features related to internet users of the most viewed YouTube videos about COVID-19 and chloroquine, during the first wave of the pandemic. We conducted a cross-sectional observational study focusing on the most viewed YouTube videos on COVID-19 and chloroquine treatment, in French and English. The primary outcome was the association between video quality as assessed by DISCERN and modified JAMA scores, and video features related to internet users as assessed by number of viewership and likes. By June 2020, 168 videos accumulating more than 57 million views and nearly 2 million reactions from Internet users, were included. Most of the videos did not support or oppose the treatment and came from news channels (N = 100; 60%). Videos taking sides were mostly pro-chloroquine (N = 69; 89%). The number of non-factual videos analyzed was very low (N = 3; 2%). The quality of the videos was average (mean DISCERN score = 2.4 (DS 1.0) and mean modified JAMA score = 2.6 (DS 0.6)) and declined over time. The best quality videos were those published by health care professionals or those from educational channels. Most experts interviewed were men (N = 136; 81%). More than 1 in 5 videos featured a political figure (N = 35; 21%), and these were mostly pro-chloroquine with lower DISCERN or modified JAMA scores (p<0.001). We found an association between the number of likes and the quality of the videos evaluated by the DISCERN score (rho = 0.29; p<0.001) and the modified JAMA score (rho = 0.30; p<0.001). The association observed between the quality of the videos and the number of likes suggests a better health literacy and critical thinking of lay internet users. Although YouTube has become a major player in the dissemination of medical information, more involvement of health professionals and governmental organizations is needed.

## Introduction

By April 2020, about half of the world's population was under some form of lockdown due to the coronavirus pandemic, leading to an increased internet traffic by around a quarter in

**Competing interests:** The authors have declared that no competing interests exist.

major cities. Seeking out medical or non-medical information, entertainment or to cope with social distancing, people stuck at home were more likely to use social media and streaming services. At the same time, publications on the coronavirus theme, whether they came from the media, health organizations and professionals, politicians or influencers grew exponentially [1, 2]. There are many reasons for this: the fear of being infected and the need for information related to the symptoms, complications, and treatment of the disease, but also the desire to inform continuously, to react and share opinions as quickly as possible, and simply to share knowledge and sources with subscribers. The consequences have been a tsunami of information, misinformation, or disinformation both online and offline, which is now qualified by the term infodemic.

Very early on, the World Health Organization (WHO) became aware of the negative consequences in terms of public health of this infodemic, fostering confusion and a feeling of mistrust towards government and scientific recommendations [3]. A collaboration between the WHO and Internet giants was set up to limit the dissemination of erroneous information [4, 5]. In December 2020, the WHO declared that the management of the COVID-19 infodemic had become as important as the management of the pandemic [6].

Among the pandemic-related issues that have ignited the web, there was the use of chloroquine as an early treatment for people infected with the coronavirus [7]. To better understand crisis communication and the public's perception risk of COVID-19 we conducted a study with the aim of recognizing the most viewed YouTube videos on chloroquine and COVID-19 during the first wave of the pandemic and to evaluate the association between the quality of the information published on YouTube and video characteristics.

## Materials and methods

### Study design

We conducted a cross-sectional observational study focusing on the most viewed YouTube videos on COVID-19 and chloroquine treatment during the first wave of the pandemic. Due to the lack of patient or public involvement, our study was exempt from any ethical review; the collection and analysis method complied with the conditions of YouTube videos that are publicly available.

### Study aims

Our primary objective is to evaluate the association between video quality as assessed by DISCERN and modified JAMA scores, and video features related to internet users as assessed by number of viewership, likes and comments. Our secondary objectives are (a) the association between quality and intrinsic features of videos as assessed by the publication category, the nature of video content and the characteristics of the expert questioned, (b) the association between featured related to internet users and intrinsic features of videos, but also (c) the evolution of video quality overtime.

### Search protocol

The YouTube search was conducted on June 4, 2020. This search was performed on the Google Chrome browser after deleting all browsing history and cookies, and after logging out of any Google account. The results were sorted using the default "view count" filter. The first 100 videos for each keyword association were selected. There was 6 keyword association, chosen to be the most representative of the searches made on this theme: "COVID19" or "Sars-Cov-2" or "Coronavirus" and "Hydroxychloroquine" or "Chloroquine" [8, 9].

The selected videos had to be presented in English or French; English being the first language adopted for scientific writing and French representing, through Professor Raoult and his position papers, an important part of the controversy on this topic [7, 10]. The exclusion criteria included duplicate videos, videos exceeding one hour in duration (based on previous studies and because graphs in video ranking systems show a popularity peak at 15 min with a decrease of views for video longer than one hour [11, 12]), videos unrelated to the theme of treatment with chloroquine in the context of COVID-19 and videos removed from YouTube after inclusion and before viewing. The hyperlinks of the first most widely viewed 100 videos were collected for each keyword (most web surfers do not look beyond the third page of page results [13]).

A first exclusion was made based on the analysis of the title and the summary of the pre-selected videos so that they corresponded to the inclusion criteria. All the retained links were then grouped together and duplicates were removed. The remaining videos formed the database. The list of analyzed YouTube videos is available in S1 File.

The impact criteria of each video (number of views, likes, dislikes and comments) were collected on the single day of June 4th, 2020, in order to ensure the transversality of the study.

## Data collection

Two reviewers (CP and YLN) independently viewed and analyzed each selected video. For each video, the number of views, likes, dislikes, and comments were collected on the same day to ensure an appropriate cross-sectional evaluation. The intrinsic video features collected were the date of publication of the video, the geographic origin, the duration, the publication category (news channels, health professionals, educational organizations, lay users, hospital/foundation, other), the nature of video content (neutral or taking position in favor or disfavor of chloroquine; factual or non-factual information with erroneous, discriminatory or racist messages), characteristics of the expert presenting the video (gender, politician). The quality of each video was assessed with the DISCERN and the modified JAMA scores. The 2 reviewers practiced on several videos to calculate the DISCERN and modified JAMA scores before starting the data collection. Both scores were developed for non-expert patients and includes questions that do not require prior scientific knowledge. We judged the form of the information provided and if the video is a useful and appropriate source of information about treatment choices; we did not judge the substance, and in our specific case whether or not chloroquine was suitable for the treatment of COVID-19.

The DISCERN score is a validated instrument to assess the quality of consumer health information [14, 15]. It includes 16 items; each item is graded from 1 (no, quality criterion not completely fulfilled) to 5 (yes, quality criterion completely fulfilled). The 16th item is the overall quality rating at the end of the instrument with the same scale of 1 to 5. We also calculated a modified DISCERN: the sum of the 15 first items of the DISCERN score in order to evaluate the better interrater agreement (DISCERN sum score). This DISCERN sum score is assessed by a scale from 15 (quality criterion not completely fulfilled) to 75 (quality criterion completely fulfilled). Our criteria for scoring the DISCERN score for YouTube videos about COVID-19 and chloroquine treatment are specified in S2 File.

The modified JAMA score is made up of 4 core standards to evaluate websites: authorship, attribution, disclosure and currency (each item scoring from 0 to 1, 0 means quality criterion not completely fulfilled and 1 means quality criterion completely fulfilled) [16]. Our criteria for scoring the modified JAMA score for YouTube videos about COVID-19 and chloroquine treatment are specified in S3 File.

Any differences in ratings regarding the characteristics of the videos were discussed secondarily between the two reviewers (CP and YLN) to reach consensus. Any discrepancies between

reviewers were resolved by discussion with the third author (RP). The complete database is available in S1 Table.

## Statistical analyses

Descriptive analyses were made for each video characteristic. The association of DISCERN and modified JAMA scores with continuous variables was assessed using Spearman rank correlation. The comparison of ordinal and count data according to video characteristics used Wilcoxon-Mann-Whitney or a Kruskal-Wallis test. Proportions were compared by $X^2$ test or Fisher's exact tests, as appropriate. Interrater agreement was assessed using the weighted kappa statistic for ordinal scores and the intraclass correlation coefficient assuming randomly selected raters for the DISCERN sum score. The evolution of DISCERN and JAMA scores over time, based on the publication date, were analyzed using a Bayesian proportional odds model, considering scores as ordinal outcomes. Given the multiple analyses, $p$ values $< 0.001$ were considered as indicating statistical significance. Analyses used The R statistical software version 3.6.3 (The R Foundation for Statistical Computing, Vienna, Austria).

## Results

A total of 168 videos were included, totaling 67.2 hours, 57,505,747 views, 894,313 likes, 791,478 dislikes and 320,060 comments (Fig 1). Videos were released between February 5[th], 2020 and May 27[th], 2020.

### Videos included

Video characteristics are presented on Table 1.

Most videos were from the United States (N = 81, 48%) or France (N = 61, 36%). The main video sources were from news media (N = 100, 60%), health care professionals or organizations (N = 29, 17%), lay users (N = 21; 12%) and educational channels (N = 15; 9%). More than one video out of 2 did not take a side with chloroquine treatment. Most of the videos taking part were pro-chloroquine (N = 69; 87%). Only 3 videos (2%) were non-factual videos. Most experts interviewed were male (N = 136, 81%). One fifth of the people speaking on chloroquine treatment were politicians (N = 35, 21%).

The mean DISCERN score was 2.4 (SD 1.0). Around one over five videos had a DISCERN score equal or greater than 4 (high quality) (n = 31; 19%). Only 3 (1,8%) videos had a score of 5. The mean modified JAMA score was 2.4 (SD 0.6). Around one over three videos had a modified JAMA score equal or greater than 3 (n = 54; 32%).

### Interrater agreement for DISCERN and modified JAMA scores

For the DISCERN score, a moderate to fair agreement was observed (weighted kappa 0.62, 95% CI 0.54 to 0.69); the agreement between the DISCERN sum scores was higher (intraclass correlation coefficient 0.84, 95% CI 0.70 to 0.91). An excellent agreement was obtained for the modified JAMA score (weighted kappa 0.91, 95% CI 0.83 to 0.96). A perfect or near-perfect (only one disagreement) was found for all modified JAMA scores items, except attribution, for which the kappa statistic was 0.86 (95% CI 0.76 to 0.93).

### Primary outcomes

The number of likes was positively associated with both the DISCERN score and the modified JAMA score of videos (rho = 0.29, $p < 0.001$ and rho = 0.30, $p < 0.001$, respectively) (Fig 2). Conversely, the numbers of views or commentaries were found neither correlated with the

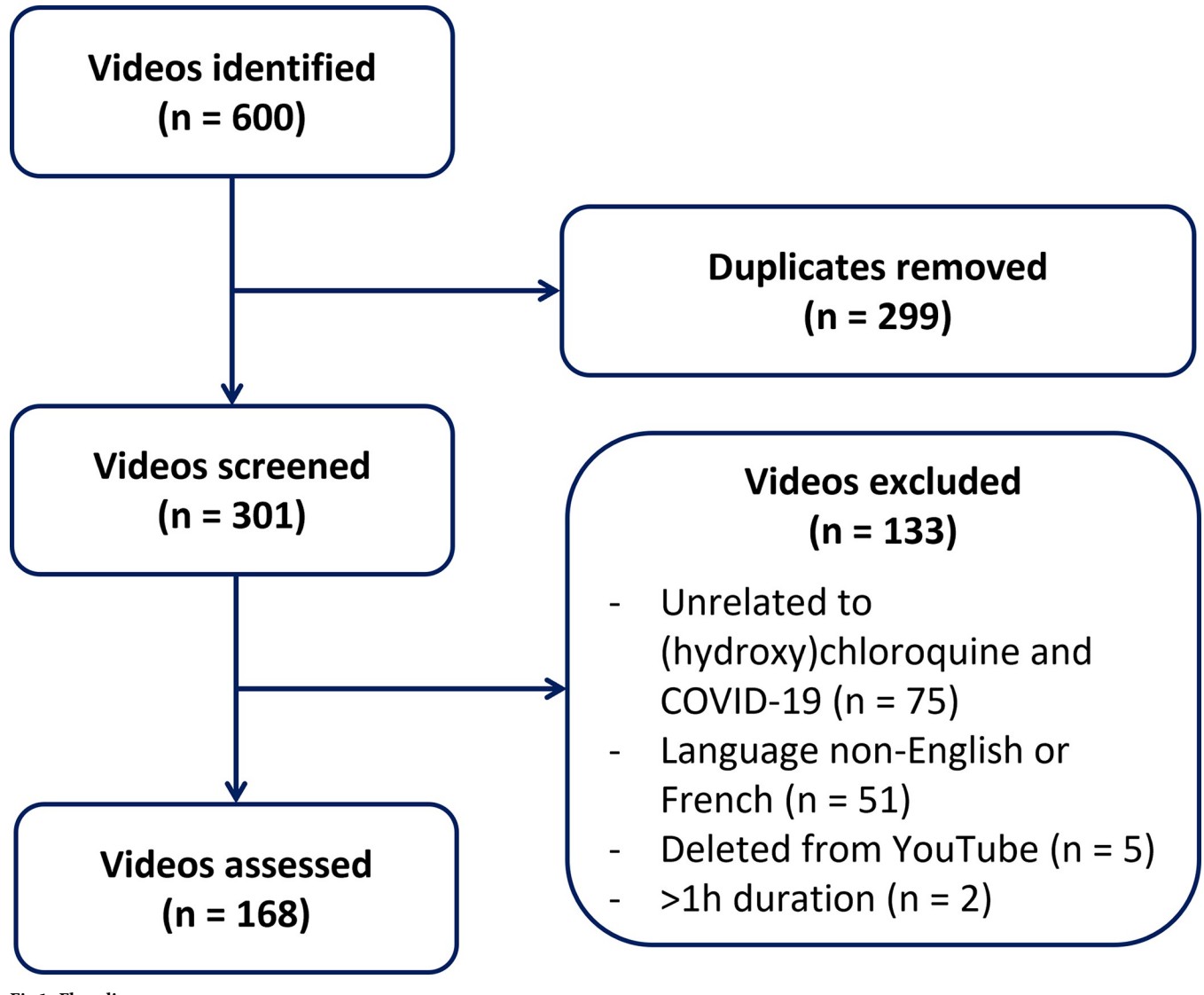

**Fig 1. Flow diagram.**

DISCERN score (Spearman's rho = 0.12, $p = 0.11$, and rho = 0.08, $p = 0.34$, respectively) or with the modified JAMA score (rho = –0.005, $p = 0.95$ and rho = 0.12, $p = 0.16$, respectively).

## Secondary outcomes

DISCERN scores of videos published by health professionals or educational channels, were greater than those published by news media or lay users ($p < 0.001$). DISCERN scores of videos published by politicians were lower than those published by others ($p < 0.001$) (Table 2).

DISCERN scores were not associated either with the geographical origin, the nature of video content or with gender of the expert (Table 2).

Modified JAMA scores of videos from educational channels were greater than those from other video categories ($p < 0.001$). Videos published by politicians had lower modified JAMA scores than those published by others ($p < 0.001$). Modified JAMA scores were not associated

**Table 1. Characteristics of the 168 videos.**

| Characteristic | No. | Value |
|---|---|---|
| **Geographical origin, no. (%)** | 168 | |
| USA | | 81 (48) |
| France | | 61 (36) |
| Asia | | 14 (8) |
| Canada | | 7 (4) |
| Europe (excluding France) | | 4 (3) |
| Africa | | 1 (1) |
| **Duration, median (IQR), min** | 168 | 6.8 (3.0–11.8) |
| **Views, median (IQR)** | 168 | 187,800 (115,963–393,819) |
| **Likes, median (IQR)** | 159 | 2,685 (1,104–5,727) |
| **Dislikes, median (IQR)** | 159 | 203 (107–532) |
| **Comments, median (IQR)** | 152 | 1,282 (548–2,404) |
| **Source, no. (%)** | 168 | |
| News channels | | 100 (60) |
| Health professionals | | 29 (17) |
| Lay users | | 21 (12) |
| Educational channels | | 15 (9) |
| Other | | 3 (2) |
| **Nature of video content, no. (%)** | 168 | |
| Neutral | | 86 (51) |
| Favorable to chloroquine | | 69 (41) |
| Unfavorable to chloroquine | | 10 (6) |
| Non-factual information provided | | 3 (2) |
| **Gender of the presenter, no. (%)** | 167 | |
| Male | | 136 (81) |
| Female | | 29 (17) |
| Both | | 2 (1) |
| **Presenter is a politician, no. (%)** | 168 | 35 (21) |
| **DISCERN score, mean (SD)** | 168 | 2.4 (1.0) |
| 1 | | 34 (20) |
| 2 | | 62 (37) |
| 3 | | 41 (24) |
| 4 | | 28 (17) |
| 5 | | 3 (2) |
| **Average DISCERN sum score, mean (SD)** | 168 | 36.2 (10.9) |
| **Modified JAMA score, mean (SD)** | 168 | 2.4 (0.6) |
| 1 | | 1 (1) |
| 2 | | 113 (67) |
| 3 | | 45 (27) |
| 4 | | 9 (5) |

either with the geographical origin, videos being factual or with the gender of the expert. The number of views was not associated with any video characteristics. The number of likes was associated with the video taking position in favor of chloroquine ($p < 0.001$). The number of comments was associated with the video taking position in disfavor of chloroquine ($p < 0.001$) (Table 2).

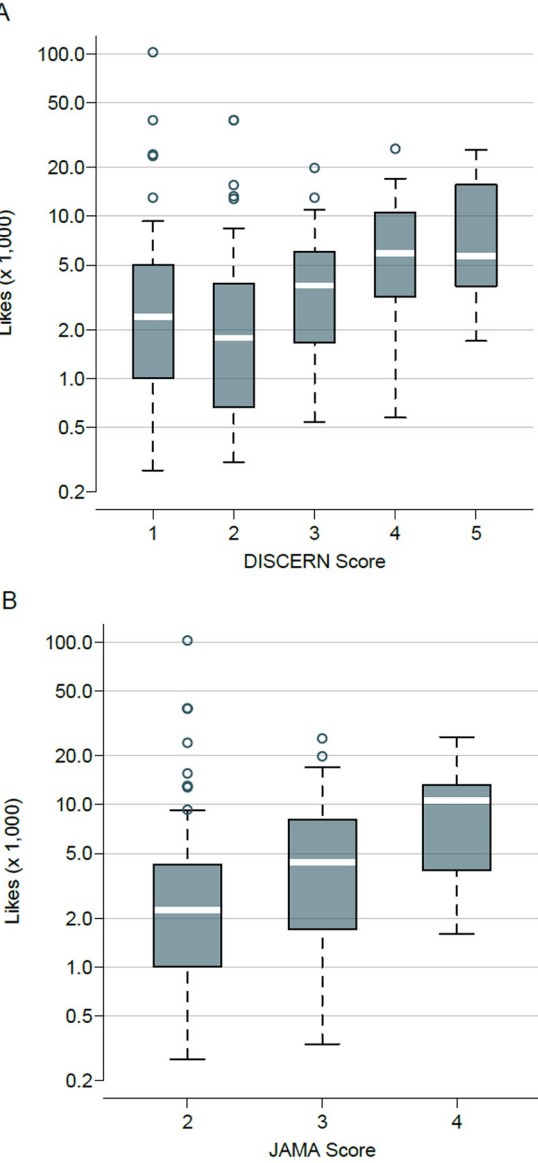

**Fig 2. Association of the number of likes with videos' quality ratings.** (A) Association of the number of likes with DISCERN score. (B) Association of the number of likes with JAMA score. Note: Number of likes was unavailable for the only video with a modified JAMA Score of 1.

Videos of politicians were favorable to chloroquine in 20/35 (57%) of cases, compared to 49/133 (37%) of videos by others ($p = 0.035$).

Videos submitted by men were more supportive of chloroquine than videos submitted by women, respectively 64/139 videos (47%) and 5/29 (17%), $p = 0,0038$ (we removed from this analysis the 2 videos presented by both sexes).

## Evolution of DISCERN and modified JAMA scores overtime

The DISCERN scores of videos published on You Tube decreased significantly over time with an OR 0.984 per day (99,9% CI 0.970 to 0.998). The modified JAMA score did not vary over time (Fig 3).

## Discussion

### Statement of principal findings

Our study confirms the importance of broadcast platforms such as YouTube for sharing fast information exchange on COVID-19. Most of the videos did not support or oppose the treatment and came from the media. The videos taking sides were mostly pro-chloroquine. The number of non-factual videos analyzed was very low. The quality of the videos was average and declined over time. The best quality videos were those published by health professionals or those from educational channels. Nearly 1 in 5 videos featured a political figure, and these were mostly pro-chloroquine. Most experts interviewed were men. We found an association between the quality of the videos and the number of likes.

### Strengths and limitations of the study

**Our study has several strengths.** We used reliable and reproducible methodology with approved quality scores. DISCERN and modified JAMA scores were used in several studies

**Table 2. Features of the 168 videos according to study variables.**

| Characteristic | DISCERN score | JAMA score | Views (x1,000)* | Likes (x1,000)* |
|---|---|---|---|---|
| **All videos** | 2.4 (1.0) | 2.4 (0.6) | 187.8 (116.0–393.8) | 2.69 (1.10–5.73) |
| **(No. assessed)** | (n = 168) | (n = 168) | (n = 168) | (n = 159) |
| **Geographical origin** | | | | |
| USA | 2.5 (1.2) | 2.3 (0.5) | 166.2 (89.6–409.0) | 3.09 (1.10–6.70) |
| France | 2.4 (0.9) | 2.4 (0.7) | 254.7 (161.5–401.0) | 3.43 (1.53–5.37) |
| Africa | 2.1 (1.0) | 2.4 (0.6) | 108.7 (69.3–163.4) | 1.00 (0.58–2.94) |
| Other | 2.6 (1.2) | 2.4 (0.7) | 178.1 (154.8–259.8) | 1.90 (1.22–3.08) |
| *P* value | 0.72 | 0.97 | 0.009 | 0.085 |
| **Source** | | | | |
| News television channel | 2.0 (0.9) | 2.2 (0.4) | 178.4 (118.7–325.9) | 2.32 (1.00–4.03) |
| Health professional | 3.0 (0.9) | 2.5 (0.7) | 172.7 (86.7–338.0) | 3.60 (1.60–5.30) |
| Lay user | 2.3 (1.0) | 2.6 (0.8) | 254.7 (119.6–437.8) | 5.00 (1.00–11.00) |
| Educational channel | 3.9 (0.5) | 3.0 (0.4) | 409.0 (141.4–693.2) | 7.83 (2.18–11.00) |
| Other | — | — | — | — |
| *P* value | <0.001 | <0.001 | 0.33 | 0.004 |
| **Factual information** | | | | |
| Neutral | 2.6 (1.1) | 2.4 (0.6) | 157.6 (92.5–287.5) | 1.70 (0.68–3.60) |
| Favorable to chloroquine | 2.4 (1.0) | 2.3 (0.6) | 260.1 (152.8–452.1) | 4.20 (2.56–7.50) |
| Unfavorable to chloroquine | 1.7 (0.9) | 2.3 (0.7) | 207.0 (108.5–369.5) | 3.85 (2.63–12.89) |
| Non-factual information | 1.0 (0.0) | 3.3 (1.2) | 437.8 (360.6–510.3) | 3.43 (2.26–13.45) |
| *P* value | 0.004 | 0.15 | 0.009 | <0.001 |
| **Gender of presenter** | | | | |
| Male | 2.5 (1.1) | 2.4 (0.6) | 211.0 (128.0–410.0) | 3.40 (1.40–6.83) |
| Female | 2.0 (0.8) | 2.2 (0.5) | 161.5 (92.3–207.4) | 1.56 (0.68–2.82) |
| *P* value | 0.012 | 0.08 | 0.020 | 0.004 |
| **Video of a politician** | | | | |
| No | 2.6 (1.0) | 2.4 (0.6) | 183.0 (100.6–394.2) | 2.76 (1.18–5.71) |
| Yes | 1.8 (0.8) | 2.1 (0.5) | 234.7 (129.5–382.0) | 2.32 (1.00–5.73) |
| *P* value | <0.001 | <0.001 | 0.51 | 0.39 |

Data are mean (SD), or median (IGR) (*). *P* values were obtained by Wilcoxon-Mann-Whitney or Kruskal-Wallis test (*)."—"indicate that these categories were not considered for comparison

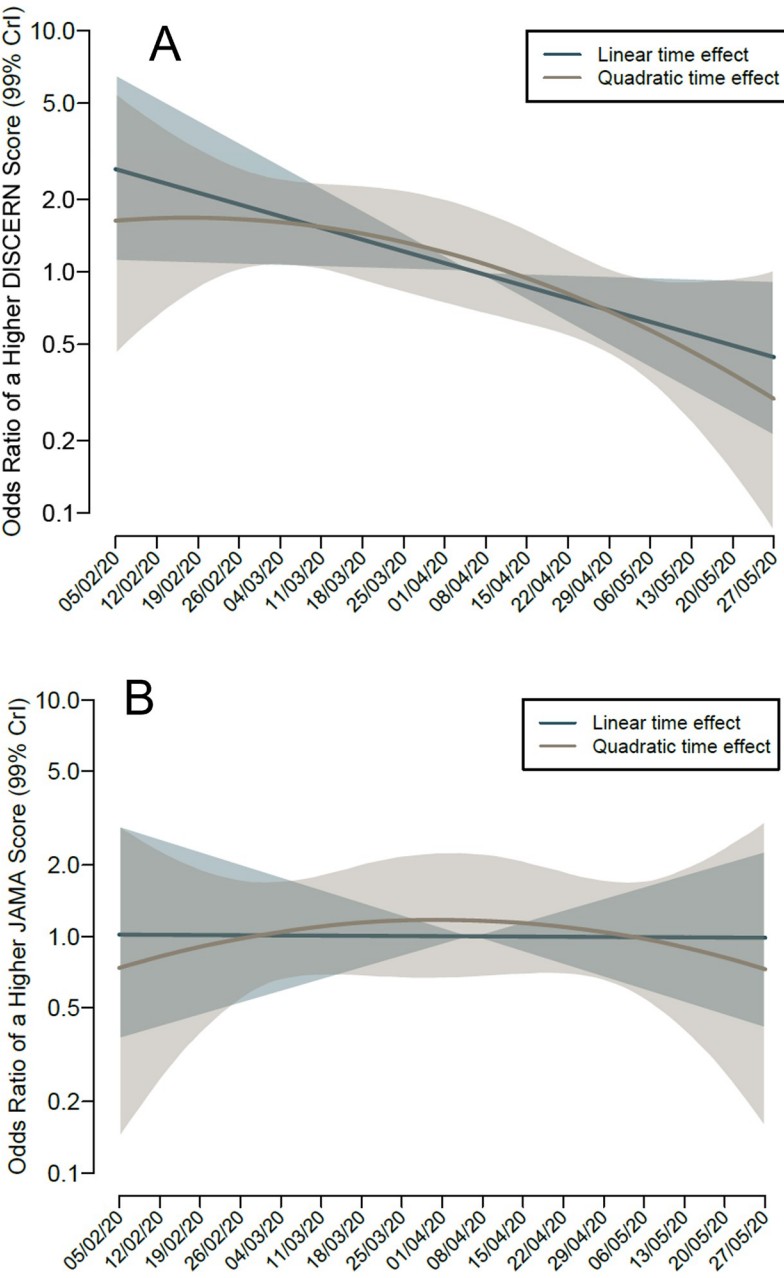

**Fig 3. Evolution of DISCERN and modified JAMA scores overtime.** (A) Evolution of DISCERN score overtime. (B) Evolution of modified JAMA score overtime.

because of their specificity for qualify medical information. Moreover, we examined 168 YouTube videos, which represent one of the biggest samples in the literature. To our knowledge, we report here the first study reviewing the quality of YouTube video regarding the use of chloroquine for the treatment of COVID-19 for the English and French languages. This subject was source of debates within medical community, and it represented a perfect example of infodemic.

**Our study also has some limitations.** As with other cross-sectional studies, our study suffers from several selection biases. We have only included videos in English and French. Our

results were limited to a given period but this period makes sense because the pro-anti-chloroquine debate dried up in May 2020, following publications suggesting the lack of benefits of chloroquine and the WHO recommended that its use be stopped by May 25[th] 2020 [17–24]. We chose to study videos published on YouTube because it is the most widely used video platform worldwide [25]. We did not study the impact that sharing these videos on other social networks may have had. The DISCERN sore interrater agreement was greater for DISCERN sum score than DISCERN score. Such discrepancy has already been described and is related to the modality to calculate the DISCERN score in comparison to the DISCERN sum that seems more precise [26].

## Meaning of the study, possible explanations, and implications

**Association between quality and likes.**   The existence of an association between the number of likes and the quality of published videos could reflect a better health literacy and critical thinking of Internet users. While earlier studies had suggested an association between the number of likes and conspiracy theories, or the absence of an association between the number of likes and video content [11], we found the opposite in our study. Among the underlying reasons, there may be better access to high quality data, on fact-checking websites and on certified website quality certification. In a review on the role of public physician in the COVID-19 era, one of them (Dr Wachter) reported a high level of knowledge health literacy and critical thinking of lay public [27].

**Importance of broadcast platforms for reducing knowledge transition gap.**   The results of this study suggest the importance of broadcast platforms such as YouTube for the dissemination of health information and policies. It has also inspired a couple of health professionals to become "public physicians" [27, 28]). Similar to other studies on COVID-19 and YouTube, the number of videos published by governmental organizations was anecdotal [13, 29]. Despite the development of access and use of the Internet, academic institutions and public health agencies do not seem to have taken the measure of the importance of risks communication and its hazard and outrage framework [30]. Risk communication strategy may include: plan carefully, accept the public as partners, be transparent and honest, speak with compassion, evaluate and reassess strategies [30]. The fact that videos taking sides were overwhelmingly pro-chloroquine is probably related to the fact that people who believed in chloroquine were more motivated to talk in the media or publish videos than people who did not believe in it [31].

**Few non-factual videos.**   In comparison to other studies published on videos on YouTube during the last public health crisis (H1N1, Ebola, Zika), we found a very small number of videos with non-factual data [11, 32, 33]. These results are similar to other studies on YouTube and COVID-19 [13, 29]. These observations could be due both to a better quality of the published videos and to the successful policy lead by the WHO with internet giants to limit the dissemination of false information or conspiracy theories that began as early as the spring of 2020 providing alerts and links to reliable websites [4, 5]. Nevertheless, the average DISCERN score observed in our cohort was quite similar to those observed in other cross sectional studies [26, 34–38]. It would be important to popularize among Internet users the existence of labels evaluating the quality of a website (still too little used) and the existence of scores such as DISCERN and modified JAMA allowing them to self-evaluate the quality of a video. Video platforms could also create an educational label in partnership with Health on the Net.

**A lot of politics and men.**   The large proportion of politicians interviewed as experts on COVID-19 treatment observed in our study reflects the politicization of the treatment. Their videos were mostly pro-chloroquine and of lower quality. Such phenomenon may be explained

by the populist tendency of part of the political class in many countries to the detriment of public health measures advocated by public health agencies. Most of the experts interviewed were men, and those were more likely to be in favor of chloroquine therapy than women. A French study carried out in the spring of 2020 by the CSA team showed that women health professionals (doctors or pharmacists), speaking as experts on COVID-19, were much less well presented than men (27% vs. 73%) [39]. Such observation may be related to the persistent gender gap observed in academic medicine promotions as well as in terms of first or last author positions [40–42].

## Conclusions

The debate on the use of chloroquine as a treatment for coronavirus contributed to the infodemic observed in the spring of 2020. We found an association between the quality of the videos and the number of likes, suggesting a better health literacy and critical thinking of lay internet users. The low proportion of non-factual videos observed in this study suggests the successful policy lead by the WHO with internet giants to limit the dissemination of false information or conspiracy theories. Although YouTube has become a major player in the dissemination of medical information, videos from medical academic institutions and governmental organizations are almost non-existent. Therefore, more involvement of health professionals and governmental organizations is needed.

## Supporting information

**S1 File. List of analyzed YouTube videos.**
(DOCX)

**S2 File. Scoring criteria of the DISCERN score for our study.**
(DOCX)

**S3 File. Scoring criteria of the modified JAMA score for our study.**
(DOCX)

**S1 Table. Database.** [1]Geographical origin: 1: France; 2: USA; 3: Suisse; 4: Belgique; 5: Canada; 6: Asie; 7: Afrique; 8: other.[2]Source: 1: hospital foundation/organization; 2: media, news channel; 3: radio station; 4: newspaper media; 5: website; 6: individual video; 7: health professional video; 8: government organization video; 9: educational video; 10: other. [3]Nature of video content: 0: not factual 1: in favor of chloroquine; 2: against chloroquine; 3: do not take part. [4]Non-factual information: 1: erroneous data; 2: racist or discriminatory comments; 3: conspiracy theories; 4: unfounded general recommendations; 5: other.[5]Gender of the presenter: 0: female; 1: male; 2: both.[6]Presenter is a politician: 0: no; 1: yes.[7]For each item of the DISCERN score: 1: quality criterion totally not fulfilled; 2: quality criterion very not fulfilled; 3: quality criterion partially not fulfilled; 4: quality criterion partially fulfilled; 5: quality criterion totally fulfilled. [8]For each item of the JAMA score: 0: quality criterion not fulfilled; 1: quality criterion fulfilled.
(XLSX)

## Author Contributions

**Conceptualization:** Cynthia Poncelet, Yên-Lan Nguyen.

**Data curation:** Cynthia Poncelet, Yên-Lan Nguyen.

**Formal analysis:** Raphaël Porcher.

**Methodology:** Cynthia Poncelet, Yên-Lan Nguyen.

**Validation:** Yên-Lan Nguyen.

**Writing – original draft:** Cynthia Poncelet, Yên-Lan Nguyen.

**Writing – review & editing:** Cynthia Poncelet, Raphaël Porcher, Yên-Lan Nguyen.

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
