## [Editor Report · Decision Letter 0]

21 Sep 2022

PONE-D-22-21772The COVID-19 and chloroquine infodemic: cross-sectional observational study of content analysis on YouTubePLOS ONE

Dear Dr. Poncelet,

Thank you for submitting your manuscript to PLOS ONE. After careful consideration, we feel that it has merit but does not fully meet PLOS ONE’s publication criteria as it currently stands. Therefore, we invite you to submit a revised version of the manuscript that addresses the points raised during the review process.

We look forward to receiving your revised manuscript.

Kind regards,

Saurav Basu, M.D.

Academic Editor

PLOS ONE

Journal Requirements:

2 .In your Methods section, please include additional information about your dataset and ensure that you have included a statement specifying whether the collection and analysis method complied with the terms and conditions for the source of the data.

"No, The funders had no role in study design, data collection and analysis, decision to publish, or preparation of the manuscript."

Additional Editor Comments:

Comments before peer review:

The methodology of the study needs more rigor. You need to explain clearly how you applied the criteria for establishing the quality of the videos using the DISCERN and modified JAMA scores. Explain for every point in the respective scale (for instance, how did you establish the reliability of the message in the videos). There is a strong possibility of your results being biased by the later findings from the Solidarity trial published in 2021. So how did you control for your own social desirability considering HCQ as a repurposed drug lacks credibility after publication of later research, that was not available to the video developers in June 2020.
---

## [Author Response · Author response to Decision Letter 0]

11 Nov 2022

Dear Reviewers and Academic Editor,

Thank you for your interest in our work. We are pleased to present our revised manuscript. 

Response to Saurav Basu, M.D. Academic Editor: 

1) You asked us to ensure that our manuscript meets PLOS ONE’s style requirements. 

We ensure that our manuscript meets PLOS ONE’s style requirements, we reworked particularly the tables and figures formatting (we upload our figures files to the PACE) and we renamed our files. 

2) You asked us to include additional information about our datas and ensure that we have included a statement specifying whether the collection and analysis method complied with the terms and conditions for the source of the data.

In our Methods section, we included additional information about our dataset. We specified that the collection and analysis method complied with the conditions of YouTube videos that are publicly available. 

3) You asked us to clarify the sources of funding and state what role the funders took in the study.

We clarified the source of funding in the cover letter: our study did not require funding for its design, conduct and manuscript preparation, and the authors received no specific funding for this work. Our institution solely covers the cost of publication. 

4) You asked us to specify where the minimal dataset can be found.

We made the list of analyzed videos (S1_File) and our database available (S1_Table) in the additional files. 

Response to Additional Editor comments: 

5) You asked us to be more rigorous on the methodology, to explain clearly how we applied the criteria for establishing the quality of the videos using the DISCERN and the modified JAMA scores, and to explain whether there may be biases in the judgement of effectiveness of chloroquine. 

We reinforced our methodology in the manuscript and with several additional files: 

- We explained how we applied the criteria of each score in a respective file (S2_File and S3_File) with several concrete examples for each item. The content of these files can be included in the manuscript directly, but this may be a bit overwhelming for the reader. 

- We specified in our Methods section that the aim of this work is to evaluate the quality of the information available in YouTube videos and to determine the factors that make a video an appropriate source of information about treatment choices. We did not judge if chloroquine was suitable for the treatment of COVID-19, and the video study was done in June 2020, before the Solidarity study was published.

We hope that we have been able to respond to the points raised during the review process, and we remain at your disposal for any further information. 

Sincerely yours,

Dr Cynthia Poncelet and Dr Yên-Lan Nguyen

---

## [Decision Letter · Decision Letter 1]

22 Feb 2023

PONE-D-22-21772R1The COVID-19 and chloroquine infodemic: cross-sectional observational study of content analysis on YouTubePLOS ONE

Dear Dr. Poncelet,

Thank you for submitting your manuscript to PLOS ONE. After careful consideration, we feel that it has merit but does not fully meet PLOS ONE’s publication criteria as it currently stands. Therefore, we invite you to submit a revised version of the manuscript that addresses the points raised during the review process.

The manuscript has undergone a review process by two reviewers who have provided their comments, which are available below. The reviewers commend the authors for their efforts in revising the manuscript in response to their previous feedback and have offered some minor suggestions to further enhance the quality of reporting.

Could you please carefully revise the manuscript to address all comments raised?

We look forward to receiving your revised manuscript.

Kind regards,

Lucinda Shen, MSc

Staff Editor

PLOS ONE

Journal Requirements:

Reviewers' comments:

Reviewer's Responses to Questions

**Comments to the Author**

1. If the authors have adequately addressed your comments raised in a previous round of review and you feel that this manuscript is now acceptable for publication, you may indicate that here to bypass the “Comments to the Author” section, enter your conflict of interest statement in the “Confidential to Editor” section, and submit your "Accept" recommendation.

Reviewer #1: (No Response)

Reviewer #2: All comments have been addressed

2. Is the manuscript technically sound, and do the data support the conclusions?

Reviewer #1: Yes

Reviewer #2: Yes

3. Has the statistical analysis been performed appropriately and rigorously? 

Reviewer #1: Yes

Reviewer #2: Yes

4. Have the authors made all data underlying the findings in their manuscript fully available?

Reviewer #1: Yes

Reviewer #2: Yes

5. Is the manuscript presented in an intelligible fashion and written in standard English?

Reviewer #1: Yes

Reviewer #2: Yes

6. Review Comments to the Author

Reviewer #1: The article presents the results of a correct investigation, developed with an adequate methodology and in a rigorous manner. Also, it is well structured and well written. Although it is more a report of research results than a scientific article, because there is little dialogue with the literature on the influence of social networks on social attitudes and behaviors, the manuscript is publishable with few modifications.

The sample is correct but the analysis is a bit limited: by focusing on a quantitative content analysis, the results are somewhat poor. Perhaps a more qualitative analysis that addressed the type of arguments for and against chloraquine in the videos and in the comments to the videos would have been more interesting.

In the "A lot of police and men" section, the information is provided that the majority of politicians' interventions are were mostly pro-chloroquine, but this information is not provided regarding the opinions of men and women. It would be interesting to know if there are differences in the opinions of men and women regarding the use of chloraquine.

Reviewer #2: The paper is well written and deals with a current and relevant theme that became more evident during the pandemic, which is the infodemic. I suggest that the work be published, but I suggest some minor adjustments in the text, as follows.

1 - In the methodology it is stated that the data collection was reality on June 4, 2020, however in the results the authors report the collection period and also make a temporal analysis, I suggest that this period be clarified in the methodology.

2 - In lines 230- 232 there is the following sentence: “To our knowledge, we report here the first study reviewing the quality of YouTube video regarding the use of chloroquine for the treatment of COVID-19.” I suggest only complementing for the English and French languages. For example, in the Portuguese language there are at least two articles with the same theme, and perhaps they exist in other languages as well.

7. PLOS authors have the option to publish the peer review history of their article (what does this mean?). If published, this will include your full peer review and any attached files.

Reviewer #1: **Yes: **Jorge Ruiz Ruiz. IESA, CSIC

Reviewer #2: No

---

## [Author Response · Author response to Decision Letter 1]

4 Apr 2023

Dear Reviewers and Academic Editor,

Thank you for your interest in our work. We are pleased to present our revised manuscript. 

Response to Journal Requirements:

1) You asked us to review our reference list to ensure that it is complete and correct.

We checked all the scientific papers in our references, and none have been retracted.

We checked all the non-scientific articles and web links: indeed, the link in reference 25 was no longer functional and we updated it. 

Response to Reviewer 1: 

2) You suggested that we complement our study with a more qualitative analysis, including the type of arguments for and against chloroquine in the videos and in the comments. 

We fully agree with you on the important value of a qualitative study on the type of arguments for and against chloroquine. 

We wanted to be able to provide a first study reviewing the quality of YouTube video regarding the use of chloroquine for the treatment of COVID-19, based on reproducible quantitative scores already used in the literature.

With these scores, we tried to judge objectively the form of the information provided and if the video is a useful and appropriate source of information about treatment choices; we did not judge the substance, and in our specific case whether or not chloroquine was suitable for the treatment of COVID-19. 

3) You suggested that it would be interesting to know if there are differences in the opinions of men and women regarding the use of chloroquine. 

We carried out this analysis and here is the result (added in the manuscript lines 203-205): 

Videos submitted by men were more supportive of chloroquine than videos submitted by women, respectively 64/139 videos (47%) and 5/29 (17%), p = 0,0038 (we removed from this analysis the 2 videos presented by both sexes). 

Response to Reviewer 2: 

4) You asked us to clarify the temporal analysis.

We conducted a YouTube search of the 100 most viewed videos on the subject on 4 June. The impact criteria of each video were collected on this single day in order to ensure the transversality of the study (the number of views, likes, dislikes, comments changes very quickly over time). In the video data collected, we looked at the publication dates of the videos. The videos were posted between 4 March and 6 May. The analysis of the evolution of the scores over time was based on the publication dates of the videos.

In the Search Protocol section, we stated that “The impact criteria of each video (number of views, likes, dislikes and comments) were collected on the single day of June 4, 2020, in order to ensure the transversality of the study.” Lines 90, 91.

In the Data Collection section, we stated that “The intrinsic video features collected were the date of publication of the video”, line 97. 

In the Statistical Analyses section, we stated that “the evolution of DISCERN and JAMA scores over time, based on the publication date, were analyzed using a Bayesian proportional odds model, considering scores as ordinal outcomes,” lines 135-137. 

In the Results section, we stated that “Videos were released between February 5th, 2020 and May 27th, 2020.” Line 144.

5) You suggested to clarify the sentence “To our knowledge, we report here the first study reviewing the quality of YouTube video regarding the use of chloroquine for the treatment of COVID-19.” Lines 235-237

We completed the sentence by specifying “for the English and French languages”. 

We hope that we have been able to respond to the points raised during the review process, and we remain at your disposal for any further information. 

Sincerely yours,

Dr Cynthia Poncelet and Dr Yên-Lan Nguyen

---

## [Decision Letter · Decision Letter 2]

29 May 2023

The COVID-19 and chloroquine infodemic: cross-sectional observational study of content analysis on YouTube

PONE-D-22-21772R2

Dear Dr. Poncelet,

We’re pleased to inform you that your manuscript has been judged scientifically suitable for publication and will be formally accepted for publication once it meets all outstanding technical requirements.

Kind regards,

Nafis Faizi, MD, MPH

Academic Editor

PLOS ONE

Additional Editor Comments (optional):

Reviewers' comments:

Reviewer's Responses to Questions

**Comments to the Author**

1. If the authors have adequately addressed your comments raised in a previous round of review and you feel that this manuscript is now acceptable for publication, you may indicate that here to bypass the “Comments to the Author” section, enter your conflict of interest statement in the “Confidential to Editor” section, and submit your "Accept" recommendation.

Reviewer #1: All comments have been addressed

2. Is the manuscript technically sound, and do the data support the conclusions?

Reviewer #1: Yes

3. Has the statistical analysis been performed appropriately and rigorously? 

Reviewer #1: Yes

4. Have the authors made all data underlying the findings in their manuscript fully available?

Reviewer #1: Yes

5. Is the manuscript presented in an intelligible fashion and written in standard English?

Reviewer #1: Yes

6. Review Comments to the Author

Reviewer #1: (No Response)

7. PLOS authors have the option to publish the peer review history of their article (what does this mean?). If published, this will include your full peer review and any attached files.

Reviewer #1: No

---

## [Editor Report · Acceptance letter]

1 Jun 2023

PONE-D-22-21772R2 

The COVID-19 and chloroquine infodemic:
Cross-sectional observational study of content analysis on YouTube 

Dear Dr. Poncelet:

I'm pleased to inform you that your manuscript has been deemed suitable for publication in PLOS ONE. Congratulations! Your manuscript is now with our production department. 

Kind regards, 

on behalf of

Dr. Nafis Faizi 

Academic Editor

PLOS ONE